# Numerical Simulation and Analysis of Droplet Drift Motion under Different Wind Speed Environments of Single-Rotor Plant Protection UAVs

**Juan Wang [1], Xiaoyi Lv [1], Bohong Wang [1], Xinguo Lan [1], Yingbin Yan [2], Shengde Chen [3,\*] and Yubin Lan [3,\*]**

[1] The College of Mechanical and Electrical Engineering, Hainan University, Haikou 570228, China
[2] Patent Examination Cooperation Guangdong Center of the Patent Office, CNIPA, Guangzhou 510555, China
[3] College of Electronic Engineering, South China Agricultural University, Guangzhou 510642, China
\* Correspondence: shengde-chen@scau.edu.cn (S.C.); ylan@scau.edu.cn (Y.L.)

**Abstract:** Unmanned aerial vehicles (UAVs) have been widely used in plant protection, and the mechanism of droplet deposition drift while spraying with the 3WQF120-12 produced by Quanfeng Aviation, a representative model of single-rotor plant protection UAVs in China, still requires more research. This study used a combination of computational fluid dynamics (CFD) and wind tunnel experiments to analyze the droplet deposition drift pattern of the 3WQF120-12 single-rotor plant protection UAV. The CFD modeling of the nozzle was confirmed to be feasible using wind tunnel experiments. Pearson correlation analysis was performed between experimental and simulated values, and multiple correlation coefficients reached above 0.89, which is a robust correlation. In this study, CFD simulations were performed to simulate the drift of UAV spray droplets under the rotor wind field and the combined effect of front and side winds. The deposition of droplets at different heights was simulated. The UAV's spraying conditions at different flight speeds, side wind magnitudes, and spraying heights were evaluated. According to the CFD simulation results of the 3WQF120-12 plant protection UAV, the recommended flight height is 1–3 m, the recommended flight speed is below 3 m/s, and the recommended ambient wind speed is within 3 m/s. The simulation results were verified by the field test, and the trend of the field experimental data and CFD simulation results are qualitatively consistent to verify the reasonableness and feasibility of the simulation's data. The simulated results were similar to the curves and spray area of the field test results at operating heights of 1.5 m and 3.5 m.

**Keywords:** plant protection UAV; aerial spraying; payloads; airflow field; droplet deposition

## 1. Introduction

Application is an important part of the plant growth and development process, and the application methods are mostly of the manual backpack type, ground machinery, and aerial application [1,2]. In the 1970s, as manned fixed-wing aircraft and helicopter applications became globally widespread, NASA and the U.S. Air Force developed a manned aircraft droplet drift prediction model based on Lagrangian equations and Gaussian models [3,4]. In China and other Southeast Asian countries, due to the small land area and complex terrain, ground machinery and fixed-wing aircraft have been difficult to apply on many occasions. China mainly used manual backpack sprayers and ground application machinery for a long period of time [5–7]. Since 2016, plant protection drones have started to play a big role in China's plant protection sector with advantages such as high efficiency, low cost, and suitability for applications in complex terrain [8–10]. According to the latest data from the China Agricultural Technology Center, the number of plant protection drones in China was about 4000 in 2016, and by 2021, the plant protection drones of professional pest control service organizations alone exceeded 120,000, with an operational area of more than 1.07 billion mu, and more than 200,000 flyers active in fields [11]; drones have become an

important supplement to traditional application methods on some occasions [12]. Under the influence of policy and demand, the agricultural unmanned aerial vehicle (UAV) market in China is expected to grow further in the next decade [13]. Distinguished by the number of rotors, plant protection UAVs can be divided into two categories: single-rotor plant protection UAVs and multi-rotor plant protection UAVs [14]. Single-rotor UAVs have the following advantages: they have a high load capacity, they have a longer stable endurance, and the wind field formed by a single rotor can blow the leaf surface to improve canopy penetration and effectively control the drifting problem of spraying chemicals [15–17]. Single-rotor UAVs are now very widely used in plant protection operations. Japan was the first country to develop plant protection UAVs, and its representative model, the Yamaha Rmax series, is a single-rotor plant protection UAV [18]. The 3WQF120-12 produced by Quanfeng Aviation is the representative model of a domestic single-rotor plant protection UAV, and its common nozzle model is Lu120-015 (Lechler Lechler Ltd., Metzingen, Germany). Researchers have published many studies on the 3WQF120-12. Wang et al. [16] experimentally studied the spraying effect of 3WQF120-12 on betel nut trees at different altitudes. Meng et al. [19] experimentally studied the crop coverage of 3WQF120-12 with its spraying droplets under different flight parameters. Chen et al. [20] evaluated the effective spray width of the 3WQF120-12 plant protection UAV through image analysis of the droplets on the acquisition card. However, there is still a lack of research regarding the mechanism of droplet deposition and drift during the spraying process of the 3WQF120-12 plant protection UAV. Due to the explosive growth in plant protection UAV applications, the need for application efficiency, application quality, and the assessment of potential risks of environmental pollution is increasingly prominent, while the simulation of droplet deposition drift started late and faces great research difficulties, and the number and accumulation of related studies is much lower than that of fixed-wing manned aircraft applications, which cannot meet the rapid growth of needs in time. The field environment where plant protection UAVs are applied is complex, with different growth characteristics of crops, and the heights of low crops and tall crops vary greatly. It is time-consuming and inefficient to study the spraying mechanism only using large-field experiments. Therefore, in this paper, we propose investigating key technologies and validation methods for the numerical simulation of droplet deposition and the drift of existing plant protection UAVs and to construct a spraying model by conducting a CFD simulation of the spraying mechanism of droplet application in the spraying process of 3WQF120-12 plant protection UAVs. The spray nozzle wind tunnel spraying model and the UAV spraying model were constructed and combined with a field test to observe the deposition pattern of droplets at different heights, ambient wind speeds, and flight parameters and to provide theoretical and data support for the movement pattern of droplets in the crop canopy.

Researchers in Western countries have constructed models for UAV spray droplet spraying after years of field trials and accumulating basic data. Among them, the FSCGB model established by Dumbauld et al. [21] using the Gaussian method is suitable for predicting long-range drift and simulating the effect of atmospheric stability, but not for studying the deposition and drift of spray droplets. The AGDISP model proposed by Teske et al. [22–24] is mainly based on the Lagrangian method for predicting the equations of the motion of spray droplets, including aircraft wake, aircraft fuselage, and environment turbulence effects generated during interactions with the environment. The AgDRIFT [25] model, developed by the American Spray Drift Research Group and other organizations based on the AGDISP model, covers conditions such as aircraft model, aircraft vortex, nozzle type, and weather factors. However, ADGISP and AgDRIFT do not have pesticide drift prediction models for small UAVs, and the two models do not apply to droplet drift and deposition for common UAV types in China.

In contrast, computational fluid dynamics (CFD) techniques can isolate the effects of variables such as wind field and particle size on droplet drift and deposition. This method can accurately capture the flow details of the spraying process, bridging the gap between the AGDISP and AgDRIFT models in UAV spraying. Zhang et al. [26] used computational

fluid dynamics (CFD) techniques to predict the velocity field and subsequent motion trajectory of droplets in the wake stream of a Painted Maid 510 G aircraft. They compared the droplet deposition with the AGDISP prediction with good agreement, which proved the correctness of the CFD model. The application of this droplet motion model was refined and gradually accepted in plant protection spraying, and CFD is currently a robust design tool in agriculture [27]. Initially, researchers used CFD techniques to simulate the spraying process of air-assisted sprayers in orchards [28,29]. In recent years, many CFD simulation applications have been devoted to investigating the effects of droplet size, wind speed, turbulence intensity, initial droplet velocity, droplet release height, and temperature on droplet displacement [30,31] and collection efficiency in wind tunnels. CFD simulations have been well used in the field of aerial spraying [32,33]. Ryan [34] performed a 3D near-field wake vortex for AT-802 air tractor simulations, and the results clearly reveal that the droplets flowing out of the aircraft wingtip vortex had a significant entrainment effect, where the droplets were lifted upward by the wingtip vortex and moved outward at the same time. Yang et al. [35] simulated the pesticide drift of the AGRAS MG-1 eight-rotor plant protection UAV spraying operation, and the results obtained have some significance for practical production.

There are many conditions affecting droplet drift, such as spraying height, flight speed, and side wind speed conditions. Wang et al. [36,37] measured UAV flight parameters by Beidou satellite positioning and conducted field trials combined with the spatial mass balance method and other methods to verify that the flight altitude, flight speed, and side wind speed conditions had significant effects on the drift of droplet deposition for UAV applications. Many researchers have studied the effects of the above factors on the spraying effect of plant protection UAVs. Yang et al. [38] analyzed the dynamic development pattern and distribution characteristics of the downwash airflow of an SLK-5 six-rotor agricultural UAV at different altitudes. They studied the droplet drift characteristics of a multi-rotor UAV. Zhang et al. [39] analyzed the four-rotor UAV's downwash flow field distribution characteristics at different flight speeds. Grant et al. [40] from the USA studied the drift characteristics of free-hovering UAV sprays in a wind tunnel at different wind speeds.

To study the pesticide spraying mechanism of the 3WQF120-12 plant protection UAV, in this study, the nozzle spraying model of the UAV was established by combining CFD and wind tunnel experiments, and the nozzle's set points and geometric parameters were added to the UAV model. Subsequently, simulations of the droplets' drift in the wind field below generated by the single-rotor plant protection UAV were performed under different spraying heights, flight speeds, and side wind conditions. The droplet particle size distribution, droplet duration variation law, and droplet deposition concentration cloud maps were obtained via the simulation to analyze the three-dimensional distribution of droplets and their variation laws. The results of the analysis were also verified in field tests.

## 2. Materials and Methods

### 2.1. CFD Modeling and Wind Tunnel Testing of the Nozzle

2.1.1. Nozzle Model Construction in CFD

In ANSYS Geometry, to establish the calculation model, the simulation calculation domain was set to a 20 m × 2 m × 1.1 m-long rectangle, and the nozzle position was 2 m in front of the model, 18 m behind, 1 m on each side, 0.5 m above, and 0.6 m below, which is the origin of the coordinates. Meshing was used to divide the tetrahedral mesh. The division model and the results are shown in Figure 1.

The fan nozzle model used a flat-fan atomizer model. The fan nozzle model is shown in Figure 2, and its setup parameters in Fluent are shown in Table 1.

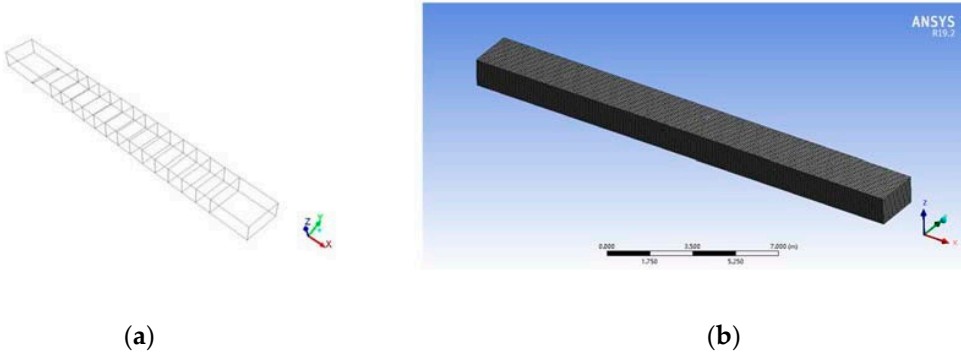

(**a**)                                                                 (**b**)

**Figure 1.** Schematic of computational domain in Fluent. (**a**) Computational domain model. (**b**) Computational domain meshing.

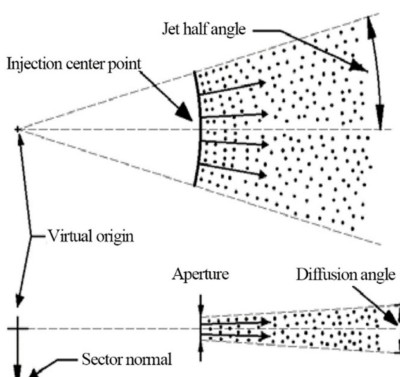

**Figure 2.** Fan nozzle spraying model.

**Table 1.** Fan nozzle parameter settings.

| Nozzle Type | Pressure Fan Nozzle |
|---|---|
| Variable name | Value |
| Coordinates of the spray center point (m) | (0, 0, 0) |
| Coordinates of the virtual center point of the jet (m) | (0, 0, 0.000115) |
| Normal coordinates of the sector surface (m) | (1, 0, 0) |
| Flow rate (L/min) | (0.59/0.68/0.56/0.92) |
| Start time (s) | 1 |
| End time (s) | 6 |
| Jet half-angle (deg) | 60 |
| Sector constants | 3 |
| Diffusion angle (deg) | 6 |

2.1.2. Nozzle Wind Tunnel Deposition Drift Test

The nozzle's wind tunnel tests were conducted in a wind tunnel laboratory at the South China Agricultural University, and the wind-tunnel-related parameters are shown in Table 2. The wind tunnel experimental arrangement is shown in Figure 3. The wind speed of this test was 0–6 m/s and the spray pressure was set to 0.3 Mpa; the specific test arrangement is shown in Table 3.

The test drift was divided into two parts. The first part was set in the direction of the spraying vertical direction of spatial drift downwind, 2 m away from the nozzle, with a total of eight collections with line spacings of 0.1 m, named V1-V8, where V1 is 0.1 m from the ground. The second part of the downwind drift test has a horizontal arrangement of 13 collection lines with a spacing of 1 m. The distance from the ground is 0.1 m, with the first horizontal collection line called H1 and the second to the fourteenth collection lines being named H3-H15, as shown in Figure 3. After spraying, the sampling lines were collected after the droplet movement was completely stabilized, and data processing was

performed. The wind tunnel experiment was arranged as follows. The spray nozzles were placed horizontally on the spray stand in the same plane as the laser emission line of the laser particle size meter, and the spray stream was perpendicular to the laser line. The wind speed in the wind tunnel was ensured to be stable and constant when testing the variation in the droplet size of the fan nozzle by examining the wind speeds, and the transmitting and receiving ends of the laser particle size meter were placed 1.5 m apart, symmetrically. The nozzle was set at 75° to the laser beam, and the distance between the nozzle and the laser beam was 0.35 m. The nozzle pressure and wind speed were set to the experimental value, the pump started to apply the spray, and the droplet size was tested and recorded after the spray stabilized. The measurement instrument was a DP-02 laser particle size meter (Omec Instruments Co., Zhuhai, China). The principle of droplet size measurement was to direct the laser beam through the droplet, due to the different sizes of the droplets, to scatter light angle changes. The receiving end was equipped with a light sensor to receive the scattered light intensity according to the distribution of the scattered light energy to calculate the particle size distribution of the measured particles and then a computer was used to obtain the droplet size distribution spectrum.

**Table 2.** Parameters of the wind tunnel.

| Main Parameters | Performance Indicators |
| --- | --- |
| Test section size (m m m) | 20 × 2.0 × 1.1 |
| Wind speed range (m/s) | 2 to 52 |
| Adjustment method | Variable frequency continuously adjustable |
| Turbulence (%) | <1.0 |
| Axial static pressure gradient | <0.01 |
| Dynamic pressure stability coefficient (%) | <1 |
| Average airflow deflection angle (°) | <1 |

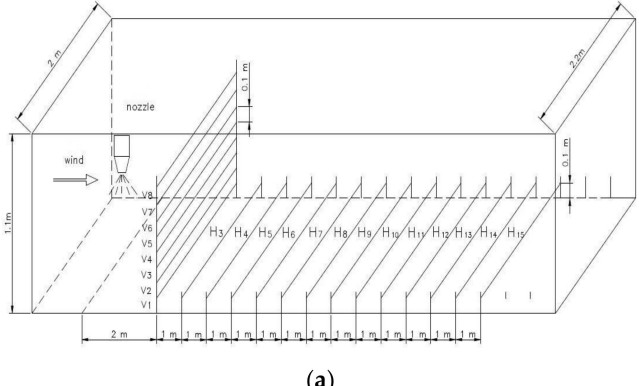

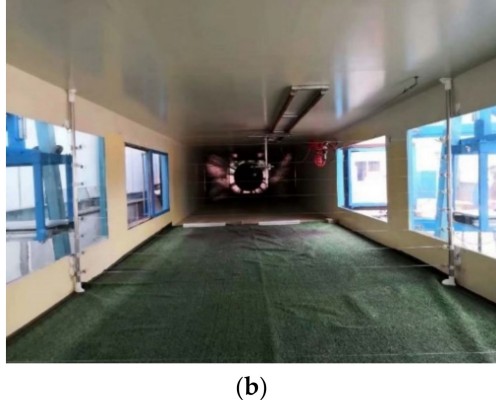

(a)          (b)

**Figure 3.** Schematic diagram of nozzle droplet drift test in the wind tunnel. (**a**) Schematic diagram of the test arrangement of droplet drift. (**b**) Sampling line arrangement test in the wind tunnel.

**Table 3.** Parameter settings for the wind tunnel drift test.

| Nozzle Type | Pressure Setting/Mpa | Flow Rate/L/min | Wind Tunnel Test Airflow Speed/m/s | Room Temperature/°C | Relative Humidity |
| --- | --- | --- | --- | --- | --- |
| Lu120-015 | 0.3 | 0.59 | 0/1/3/6 | 28 | 65% |

In testing whether the fan nozzle droplet drift was affected by the wind speed, the spray nozzle's spray surface was adjusted to be perpendicular to the wind speed of the wind tunnel. The spraying liquid was an aqueous solution of rhodamine-B added at a mass fraction of 5‰, and the rhodamine tracer was a violet-red soluble fluorescent tracer, with droplet drift and deposition collected by polyethylene lines. The test drift was divided

into two parts: the first part was set in the downwind direction of the spray application of vertical spatial drifts, 2 m away from the nozzle, with a total of eight collection lines with a spacing of 0.1 m, named V1-V8, where V1 was 0.1 m from the ground. The second part of the downwind drift test was the horizontal arrangement of 13 collection lines with a spacing of 1 m, where the distance from the ground was 0.1 m and the first horizontal collection line was called H1 and the second to the fourteenth collection lines were named H3-H15, as shown in Figure 3. After the end of the test spraying and after the droplet movement was completely stable, the droplets on the collection line were fully dried and the test personnel from the observation window and air outlet into the wind tunnel put on disposable rubber gloves. Then, a wand set of electric drill uniform collection sampling lines and a fluorescence spectrophotometer F-380 (Guangdong Technology Development Co., Tianjin, China) were used to measure the experimental value.

## 2.2. Geometric Model Construction of 3WQF120-12 Single-Rotor UAV

The main parameters of the 3WQF120-12 single-rotor plant protection UAV used in this study are shown in Table 4, and the physical diagram is shown in Figure 4a.

**Table 4.** Parameters of the 3WQF120-12 UAV.

| Name | Global Eagle |
|---|---|
| Number of nozzles (pcs) | 3 |
| Nozzle type | Lu120-015 |
| Main rotor speed (rpm, RPM) | 1300 |
| Tail rotational speed (rpm, RPM) | 5950 |
| Medicine box capacity (L) | 12 |
| Take-off weight (kg) | 47 |
| Main rotor diameter (m) | 2.41 |
| Body size (L*W*H, m) | $2.13 \times 0.70 \times 0.67$ |

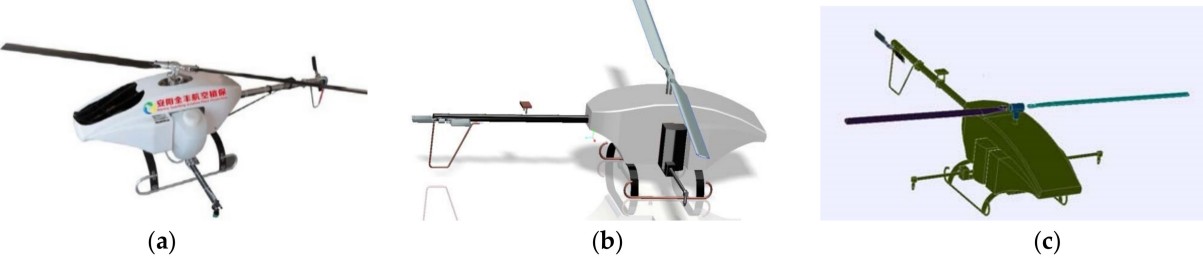

(**a**)                                                                   (**b**)                                                                   (**c**)

**Figure 4.** A 3WQF120-12-type plant protection UAV physical and model diagram. (**a**) Physical drawing. (**b**) Airframe model and complete assembly model. (**c**) Mesh division results.

The UAV model is divided into a fuselage part and a rotor part. The fuselage part was modeled by measuring the corresponding data and using Pro/Engineer software. The main rotor and tail wing were scanned by a GOM 3D scanner. Firstly, reflective enhancers were sprayed on the main rotor and tail wing parts, and marker points were pasted onto the relevant parts and then scanned. The scanned point cloud file was imported into Geomagic Design X software, the 3D solid model was constructed by reversing the point cloud file, the rotor part was assembled with the fuselage part in Pro/E software, and the model was output in. stp format; the model is shown in Figure 4b.

In ANSYS, the fluid computational domain was created and suppressed by Boolean operations. The fluid computational domain included the external flow field computational domain, the main rotor rotation domain, and the tail rotor rotation domain. The size of the external flow field computational domain was 14 m × 14 m × 14 m. The center of the main rotor is the center of the UAV, and the distance from the left, front, and upper sides is 4 m. The computational domain model file was imported into Simlab software for meshing, and the meshing results are shown in Figure 4c.

### 2.3. Computational Model of Rotor and Droplet Motion in CFD

2.3.1. CFD in the Rotor Wind Field Calculation Model

In this study, the pressure-based transient simulation was solved as a numerical simulation model, and the gravity magnitude was set to 9.81 m/s$^2$. The momentum, mass, energy, and other passive scalar quantities in fluid motion were mainly transmitted by large vortices, which better reflect the turbulence problem. The large eddy simulation (LES) model was chosen as the turbulence model to accurately simulate the rotor flow field distribution and to capture the details of the surrounding airflow field during the spraying of a single-rotor UAV. The subgrid-scale turbulence model in ANSYS Fluent was used to calculate the subgrid turbulent stresses.

$$\tau_{ij} - \frac{1}{3}\tau_{kk}\delta_{ij} = -2\mu_t \bar{S}_{ij} \tag{1}$$

where $\mu_t$ is the turbulent viscosity at the subgrid-scale. Subgrid-scale stress $\tau_{kk}$ is not modeled but is added to the filtered static pressure term. $\bar{S}_{ij}$ is defined as the analytic scaled strain rate tensor.

Assuming that N is the number of vortex points and A is the area of the entrance section, the amount of vortex carried by a given particle *i* is given by the circulation $\Gamma_i$, and the assumed spatial distribution $\eta$ is expressed as follows:

$$\Gamma_i(x,y) = 4\sqrt{\frac{\pi A k(x,y)}{3N[2\ln(3) - 3\ln(2)]}} \tag{2}$$

$$\eta\left(\vec{x}\right) = \frac{1}{2\pi\delta^2}\left(2e^{1|x|^2/2\sigma^2} - 1\right)2e^{-|x|^2/2\sigma^2} \tag{3}$$

where *k* is the turbulent kinetic energy. This parameter provides control over the size of the vortex particles.

2.3.2. CFD Computational Model of Droplet Motion

The simulated single-rotor UAV spray in this paper was performed by using the Euler–Lagrange method proposed by C.T. Crowe and L.D. Smoot. Equation (4) is the discrete-phase particle equation of motion, which is expressed as follows:

$$\frac{d\vec{u}_p}{dt} = \frac{\vec{u} - \vec{u}_p}{\tau_r} + \frac{\vec{g}\left(\rho_p - \rho\right)}{\rho_p} + \vec{F} \tag{4}$$

where $\frac{\vec{u} - \vec{u}_p}{\tau_r}$ is the drag force per unit particle mass and $\vec{F}$ is the additional acceleration (force/unit particle mass).

The additional terms consider the rotation of the reference system, the temperature gradient, the Brownian motion of tiny particles, and the lift due to shear; the Magnus lift $\tau_r$ is the droplet or particle relaxation time.

$$\tau_r = \frac{\rho_p d_p^2}{18\mu}\frac{24}{C_d Re} \tag{5}$$

$\vec{u}$ is the continuous phase velocity, m/s. $\vec{u}_p$ is the particle velocity, m/s. $\rho_p$ is the particle density, kg/m3. $d_p$ is the particle diameter, and m. $\vec{g}$ is the acceleration of gravity, m/s$^2$. $C_d$ is the traction coefficient. $\mu$ is the fluid molecular viscosity. $\rho$ is the density of the fluid and $d_p$ is the particle diameter. *Re* is the Reynolds number.

$$Re = \frac{\rho d_p \left|\vec{u} - \vec{u}_p\right|}{\mu} \tag{6}$$

In this study, the TAB model in ANSYS Fluent was chosen as the droplet break-up model. Due to the low Weber number, the TAB model is well suited for low-velocity sprays in a standard atmosphere. The TAB model is a classical method for calculating droplet break-up, and it applies to many engineering sprays. The method is based on Taylor's analogy between an oscillating deformed droplet and a spring–mass system.

## 3. Conclusion and Discussion

### 3.1. Simulation Results and Experimental Validation of Spray Nozzle Application in CFD

3.1.1. Simulation Results of Wind Tunnel Spraying in CFD

Figure 5 shows the cloud diagram of droplet deposition obtained when changing the pre-wind wind speed to 0, 1, 3, and 6 m/s. The variation in the droplet deposition concentration with wind speed in the horizontal and vertical planes after 5 s of droplet spray application is shown, and the horizontal and vertical plane positions are the same as in the wind tunnel measurements.

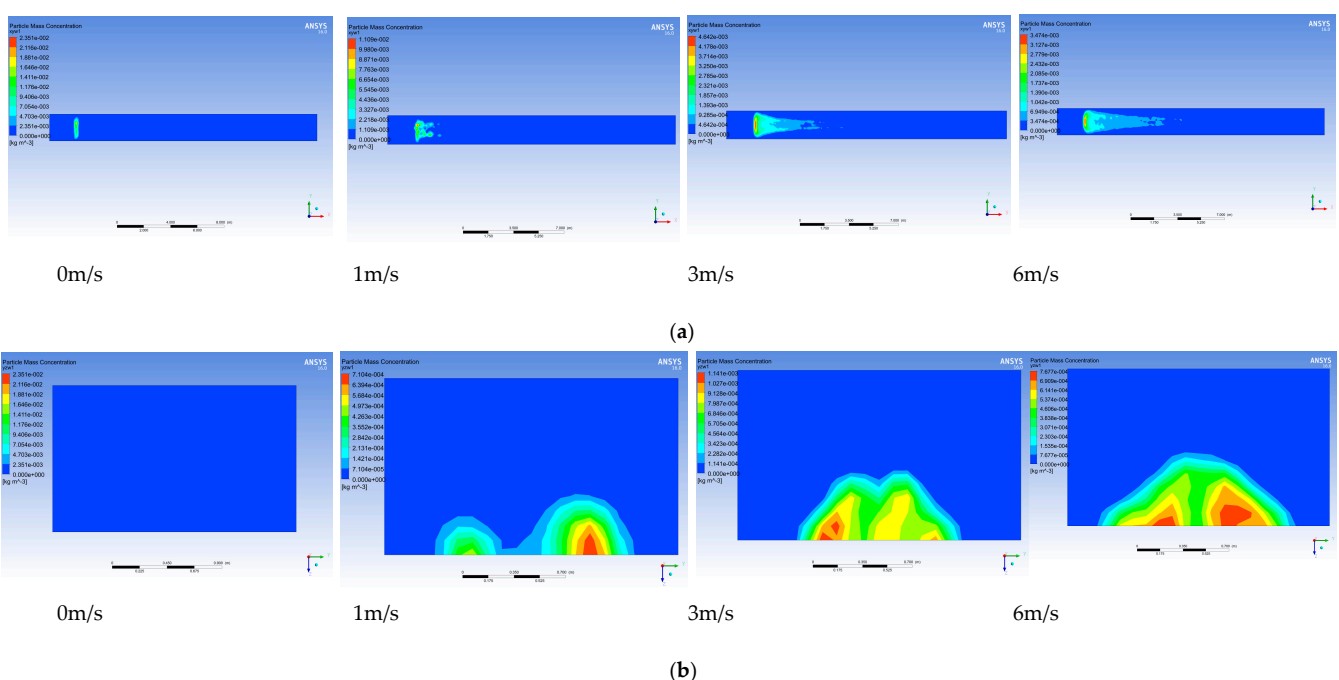

**Figure 5.** The deposition distribution clouds of the Lu120-015 spray nozzle at 0.3 Mpa in horizontal and vertical planes at different wind speeds. (**a**) The deposition distribution of the spray nozzle on the horizontal surface. (**b**) The deposition distribution of droplets on the vertical surface. Note: The lower part of the image is marked with the test wind speed.

As can be seen in Figure 5, with a wind speed of 0 m/s, a vertical plane without droplet deposition, and a horizontal droplet deposition area for a long oval type, when the wind speed increases, the vertical plane within the droplet deposition concentration area height gradually increases and the regional shape from the two sides gradually spreads to the middle; the reason for this is the shape of the fan spray surface and the direction of the incoming wind. When the wind speed is low, the droplets at the edge of the fan surface are more likely to be affected by the incoming wind, and the droplet drift area gradually becomes mountainous. The horizontal droplet deposition area increases with the incoming wind speed, droplets from both sides first begin to move downwind, the droplet deposition concentration is gradually reduced, and the deposition of the largest concentration is in locations that are still below the nozzle, gradually decreasing in value.

At a wind speed of 0 m/s, the bottom droplet deposition concentration is concentrated in the elliptical region below the nozzle and gradually decreases from the inside out. At a wind speed of 1 m/s, the droplets are affected by the incoming wind, and the

droplets can drift as far as 1.85 m. The downwind deposition region begins to bifurcate into two symmetric parts, which is due to the fan-shaped surface of the jet blocking the movement of the wind, and wind speeds gradually decrease from the left to the middle at a wind speed of 3 m/s. The deposition concentration value then decreases again, and the downwind deposition region bifurcates into two symmetric parts. At a wind speed of 6 m/s, the deposition area expands to 8.15 m and forms an obvious arc on both sides, and the deposition concentration value gradually decreases, which is consistent with the wind tunnel's test results.

### 3.1.2. CFD in the Nozzle Simulation Results and Wind Tunnel Test Comparison

To verify the accuracy of the nozzle's modeling, the CFD simulation's results were compared with data from wind tunnel tests. The data on the droplet deposition concentration of the sampled line at the same position in the CFD computational domain as in the wind tunnel were extracted. The deposition concentration indicates the mass of droplets deposited per unit area, and the simulation value indicates the value of the deposition concentration obtained by using the fluid simulation software ANSYS Fluent. Since the unit of droplet deposition concentration is $g/m^3$ in CFD and $\mu g/cm^2$ in the test, the vertical coordinate in the figure has no unit. Figure 6 shows a comparison plot of the simulated mean and experimental values.

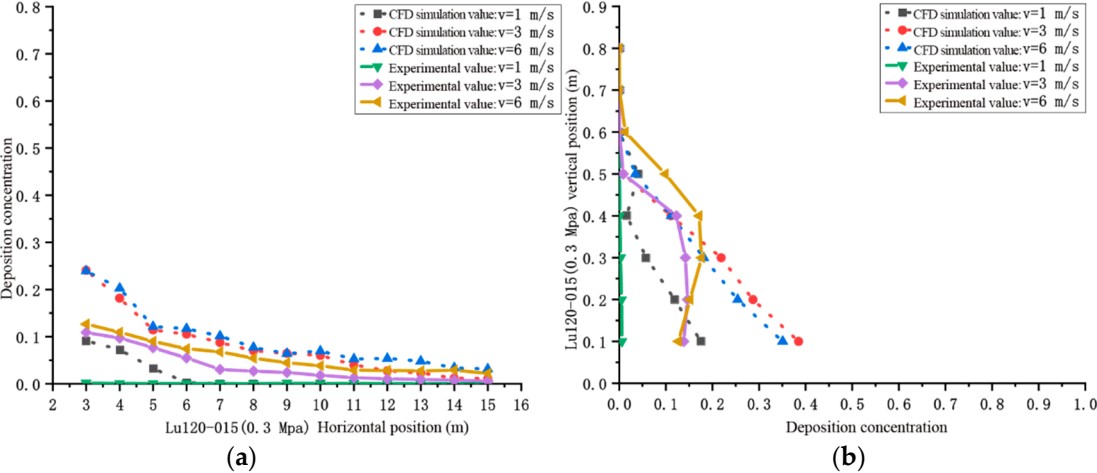

**Figure 6.** Comparison of CFD simulated droplet deposition concentration results with experimental values in horizontal and vertical planes at different wind speeds. (**a**) Comparison of the simulated droplet deposition concentration results with the experimental values on the horizontal plane. (**b**) Comparison of the simulated droplet deposition concentration results with the experimental values in the vertical plane.

From Figure 6, we can see that the simulated values in the horizontal plane have similar trends with the experimental values, and the degree of similarity is good. We also present a correlation analysis of simulated and experimental values. The simulated data in the horizontal and vertical planes were fitted at different wind speeds. The fitting equation is y = a × x$b$, and the fitting results are shown in Table 5. The simulated values of the horizontal and vertical surfaces are fitted with the linear experimental values at different wind speeds. The fitting equation is y = a + bx, and the fitting results are shown in Table 6.

**Table 5.** Fitting equation $y = a \times x^b$ for simulated values of droplet deposition at different sampling locations in the horizontal and vertical planes.

| | Simulated Wind Speed Conditions | a | b | R (Correlation) | $R^2$ | RSS/dof | RSS |
|---|---|---|---|---|---|---|---|
| Horizontal plane | v = 1 m/s | 1.70976 | −2.58746 | 0.969 | 0.896 | $5.384 \times 10^{-4}$ | $3.230 \times 10^{-3}$ |
| | v = 3 m/s | 1.11594 | −1.36459 | 0.925 | 0.972 | $5.000 \times 10^{-3}$ | $3.002 \times 10^{-2}$ |
| | v = 6 m/s | 0.88518 | −1.15306 | 0.920 | 0.974 | $3.700 \times 10^{-3}$ | $1.899 \times 10^{-2}$ |
| Vertical plane | v = 1 m/s | 0.01223 | −1.181 | 0.944 | 0.889 | $1.096 \times 10^{-4}$ | $1.210 \times 10^{-3}$ |
| | v = 3 m/s | 0.03719 | −1.05395 | 0.928 | 0.816 | $1.441 \times 10^{-4}$ | $1.590 \times 10^{-3}$ |
| | v = 6 m/s | 0.03484 | −1.03816 | 0.925 | 0.849 | $1.164 \times 10^{-4}$ | $1.280 \times 10^{-3}$ |

Note: y is the simulated value and x is the horizontal line position or vertical line sampling position; RSS/dof is the mean square of residual; RSS is the residual sum of squares.

**Table 6.** Linear fitting equation y = a + bx for simulated and wind tunnel experimental values of droplet deposition in horizontal and vertical planes.

| | Simulated Wind Speed Conditions | Test Wind Speed Conditions | a | b | R (Pearson Correlation Coefficient) | $R^2$ | RSS/dof | RSS | RMSE |
|---|---|---|---|---|---|---|---|---|---|
| Horizontal plane | v = 1 m/s | v = 1 m/s | 0.000394 | 0.00797 | 0.606 | 0.368 | $5.506 \times 10^{-7}$ | $3.303 \times 10^{-6}$ | $7.420 \times 10^{-4}$ |
| | v = 3 m/s | v = 3 m/s | −0.00322 | 0.50591 | 0.967 | 0.935 | $1.220 \times 10^{-3}$ | $7.300 \times 10^{-3}$ | $3.488 \times 10^{-2}$ |
| | v = 6 m/s | v = 6 m/s | 0.00839 | 0.52237 | 0.975 | 0.950 | $3.360 \times 10^{-3}$ | $2.018 \times 10^{-2}$ | $5.800 \times 10^{-2}$ |
| Horizontal plane | v = 1 m/s | v = 1 m/s | 0.000086395 | 0.03513 | 0.957 | 0.916 | $1.151 \times 10^{-7}$ | $1.266 \times 10^{-6}$ | $3.392 \times 10^{-4}$ |
| | v = 3 m/s | v = 3 m/s | 0.01604 | 0.42592 | 0.896 | 0.802 | $8.999 \times 10^{-5}$ | $9.899 \times 10^{-4}$ | $9.490 \times 10^{-3}$ |
| | v = 6 m/s | v = 6 m/s | 0.0445 | 0.411 | 0.716 | 0.512 | $6.452 \times 10^{-5}$ | $7.097 \times 10^{-4}$ | $8.030 \times 10^{-3}$ |

Note: y is the experimental value, and x is the simulated value; RSS/dof is the mean square of the residuals; RSS is the sum of squares of the residuals; and RMSE is the root mean squared error.

We used the Levenberg–Marquart optimization algorithm and allometric model.

From Table 5, it can be seen that the correlation coefficients of the fitted equations of the simulated data in the horizontal and vertical planes at different wind speeds are above 0.9, and the amount of droplet deposition is exponentially related to the sampling location. From Table 6, it can be seen that in the horizontal plane sampling analysis, with the exception of Lu120-015 at 1 m/s, the correlation between the experimental and simulated values is low because the wind speed is small, the deposition volume is close to 0 after 3 m, and the residual sum of squares (RSS) value is also low; moreover, the data error is small. All other groups show an extremely strong correlation between the experimental and simulated values. In the vertical plane, the correlation coefficient is strong at a wind speed of 6 m/s, and all other groups exhibited extremely strong correlations. The comparative data demonstrate that the simulation of droplet spraying with CFD nozzles is more reliable and can qualitatively analyze the droplet deposition distribution, which is a feasible basis for the next step of constructing UAV spraying models.

### 3.2. CFD in UAV Spraying Simulation Results

3.2.1. Simulation Results of Droplet Size and Duration of UAV Spraying

Set points and geometric parameters of the Lu120-015 nozzle were added to the 3WQF120-12 single-rotor UAV model. We simulated the spraying of droplets under the influence of the rotor wind field, forward wind, and side wind. The simulated UAV flight speeds were 0, 3, and 6 m/s, and the side wind speeds were 0, 3, and 6 m/s. The time step was set to 0.001 s. After the rotor wind field was calculated for 10 s to reach stability, the spraying started for 5 s according to the initial settings to obtain the duration and particle size distribution of the droplets under the rotor wind field. The different velocity combinations in the experiment are expressed as n-m, where n denotes the front wind speed, which represents the UAV flight speed, and m denotes the side wind speed. Figure 7 shows the particle size distribution of droplets under different wind speed conditions. Figure 8 shows the distribution of the droplet size at different wind speeds. The front view is on the left and the side view is on the right.

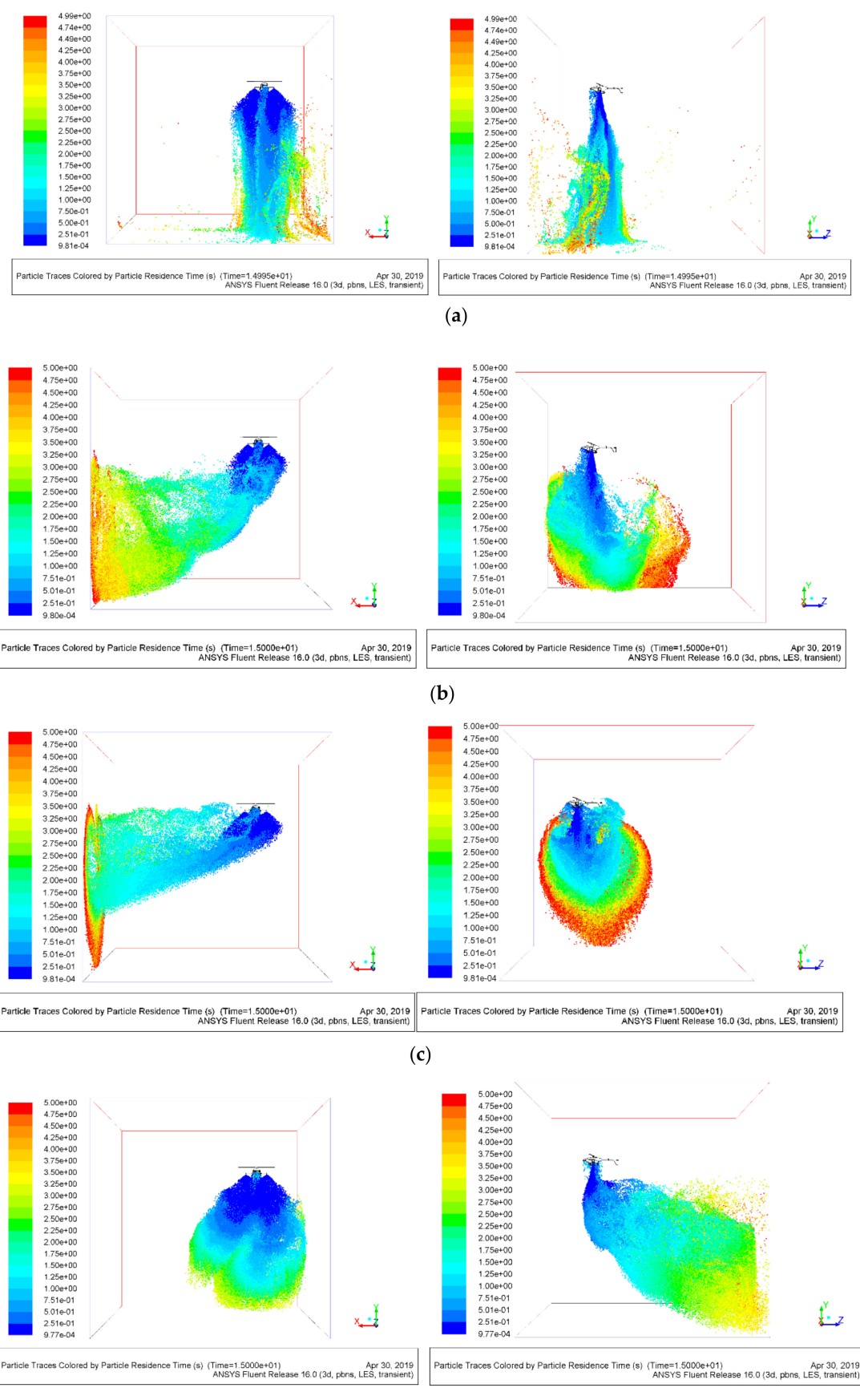

**Figure 7.** *Cont.*

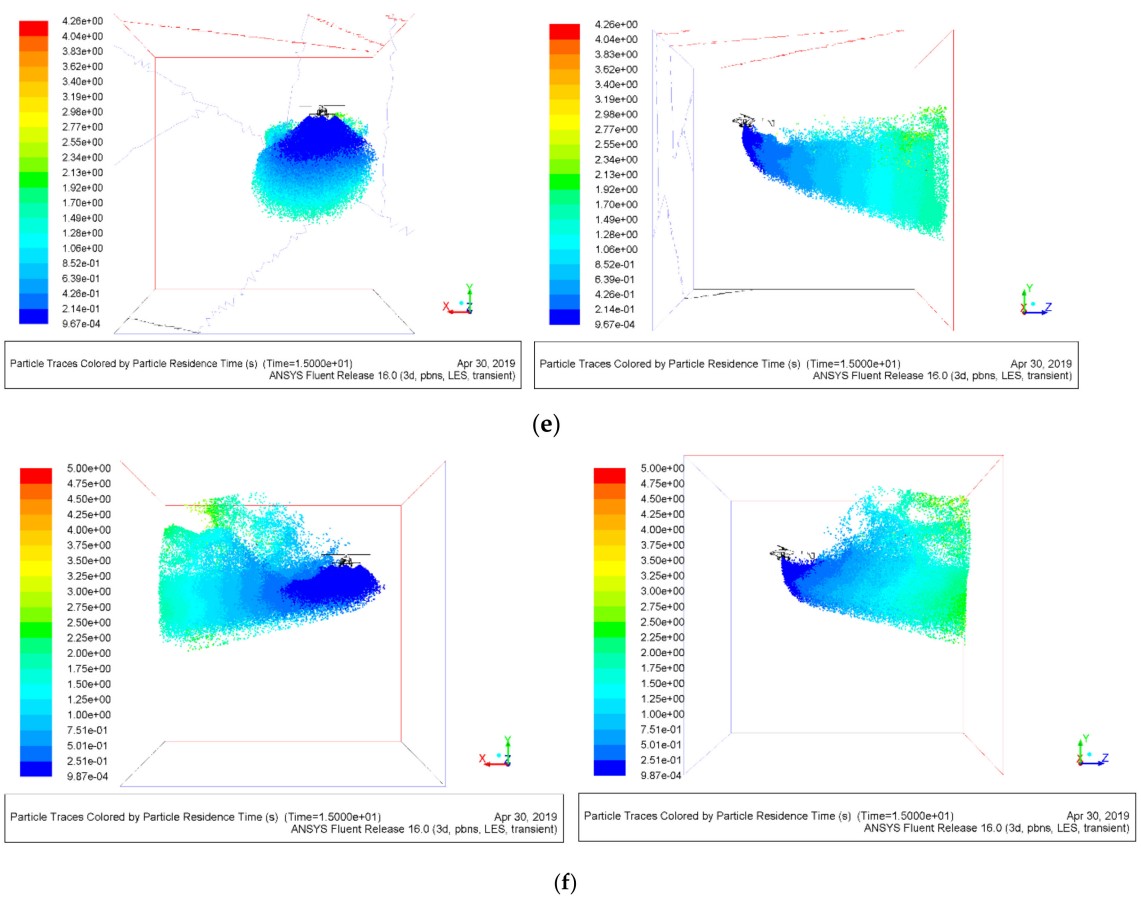

**Figure 7.** Distribution of droplet duration under different combinations of wind fields: (**a**) 0–0. (**b**) 0–3. (**c**) 0–6. (**d**) 3–0. (**e**) 6–0. (**f**) 3–3. Note: The front view is on the left and the side view is on the right. The different velocity combinations in the experiment are expressed as n-m, where n denotes the UAV flight speed and m denotes the side wind speed.

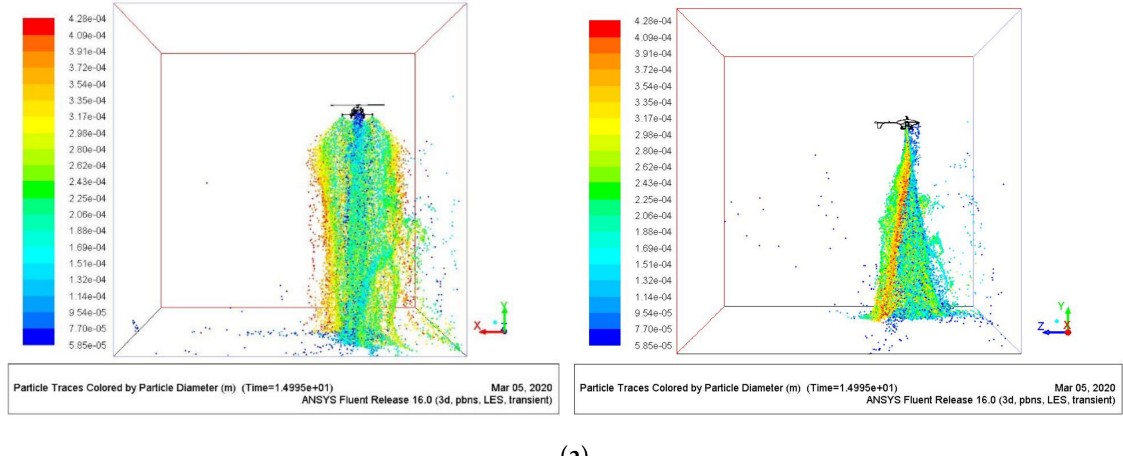

(**a**)

**Figure 8.** *Cont.*

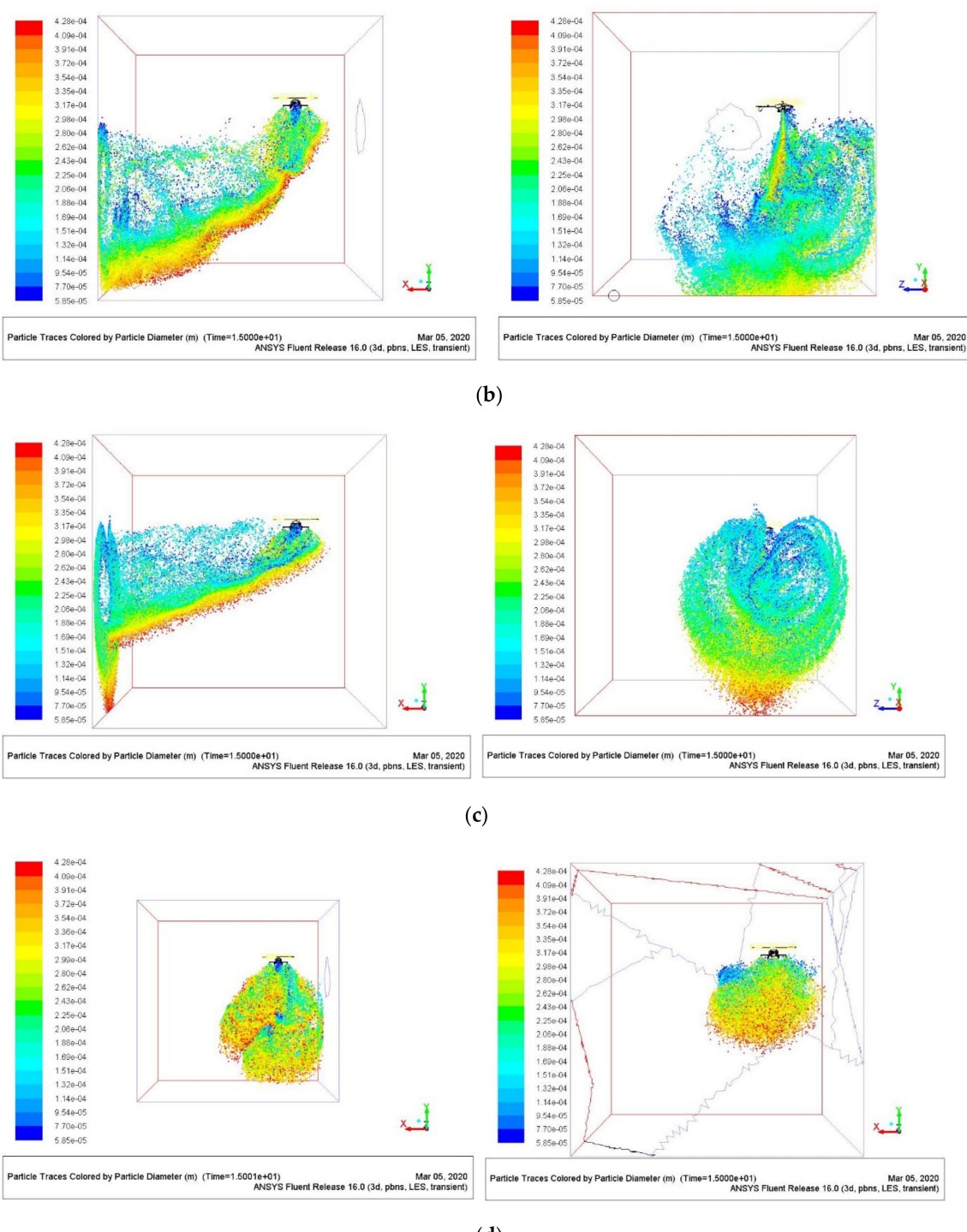

**Figure 8.** *Cont.*

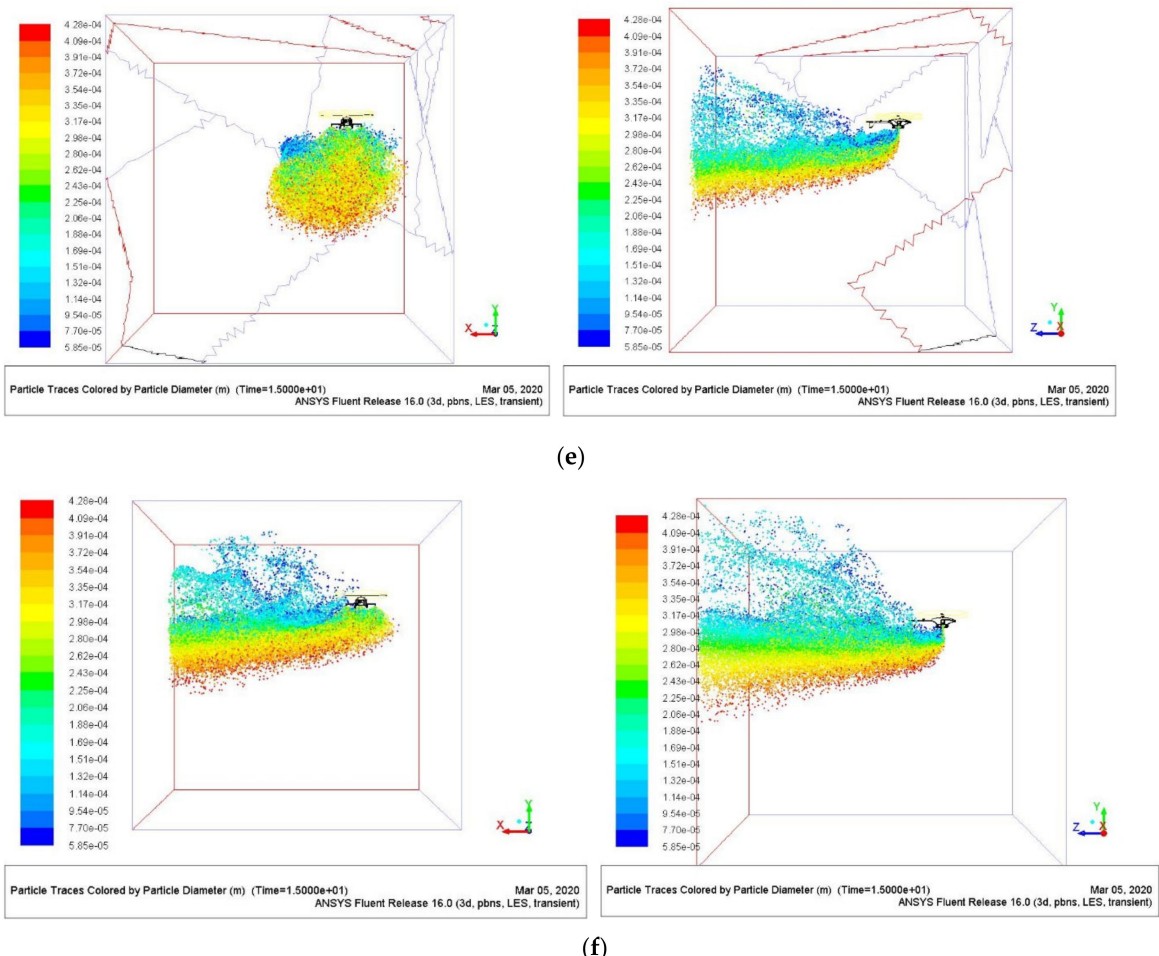

**Figure 8.** Distribution of droplet size under different combinations of wind fields: (**a**) 0–0. (**b**) 0–3. (**c**) 0–6. (**d**) 3–0. (**e**) 6–0. (**f**) 3–3. Note: The front view is on the left and the side view is on the right. The different velocity combinations in the experiment are expressed as n-m, where n denotes the UAV flight speed and m denotes the side wind speed.

In Figure 7a–c, the front wind speed is 0 m/s, i.e., when the UAV is hovering, the droplet motion is only observed by changing the size of the side wind. Figure 7a shows the droplet particle size motion time distribution when the side wind is 0 m/s; most particles with a longer motion time are concentrated in a small area on the right side under the rotor. When the side wind speed increases to 3 m/s, some droplets moved to the boundary of the calculation area due to the influence of the side wind. When the sidewind speed continues to increase to 6 m/s, the range of short-time moving droplets is further reduced, and a large number of long-time moving particles are gathered at the boundary of the calculation domain, resulting in droplet drift. The other treatments are similar to the above discussion when the flight speed is constant and the side wind is changed, so we will not discuss them here.

The main views of Figure 7b (0–3), Figure 7d (3–0), and Figure 7f (3–3) show that the droplet particle duration range is further reduced under the combined effect of the front and side winds, and the droplets with a duration of 2.5 s or less are retained in the computational domain, while the rest of the droplets have escaped from the computational domain. Figure 7f shows that no droplets exist beyond 2 m directly below the rotor, and this operational parameter is not favorable for crops larger than 1 m in height. Figure 7f shows that the droplet particles' movement range becomes larger and the shape is irregular, causing the spraying area to deviate from the course, which is mainly attributed to the influ-

ence of the side wind, resulting in a change in the spraying area, and care should be taken to prevent re-spraying and leakage in the practical application of this operating parameter.

Combined with the simulation results of the droplets' size, we can see that the largest proportion of 0–0 droplets is in the range of 250–280 μm, and it is mainly concentrated in the position below the rotor. The fine droplets below 100 μm tend to be interspersed in the center of the droplet group under the effect of the rotor's wind field. When the side wind speed increases to 3 m/s, the droplets with a particle size of less than 150 μm are more likely to be affected by airflow and float in the air. The fine droplets take a longer time to move in the calculation domain, and some of them escape from the boundary of the calculation domain after 5 s of movement. When the side wind increases to 6 m/s, the drifting phenomenon is more obvious, and it is worth noting that a large proportion of the escaped droplets are smaller than 250 μm, and some of the fine droplets only need 2 s to escape from the computational domain.

### 3.2.2. Simulation Results of the Cloud Map of Droplet Deposition Concentration at Different Heights

Figure 9 shows the cloud plots of droplet deposition concentration at different height planes under different wind speed conditions, and for each wind speed condition, the cloud plots of droplet deposition concentration at 1, 3, 6, and 10 m height are shown in order. The change of color from blue to red indicates the change in droplet deposition concentration.

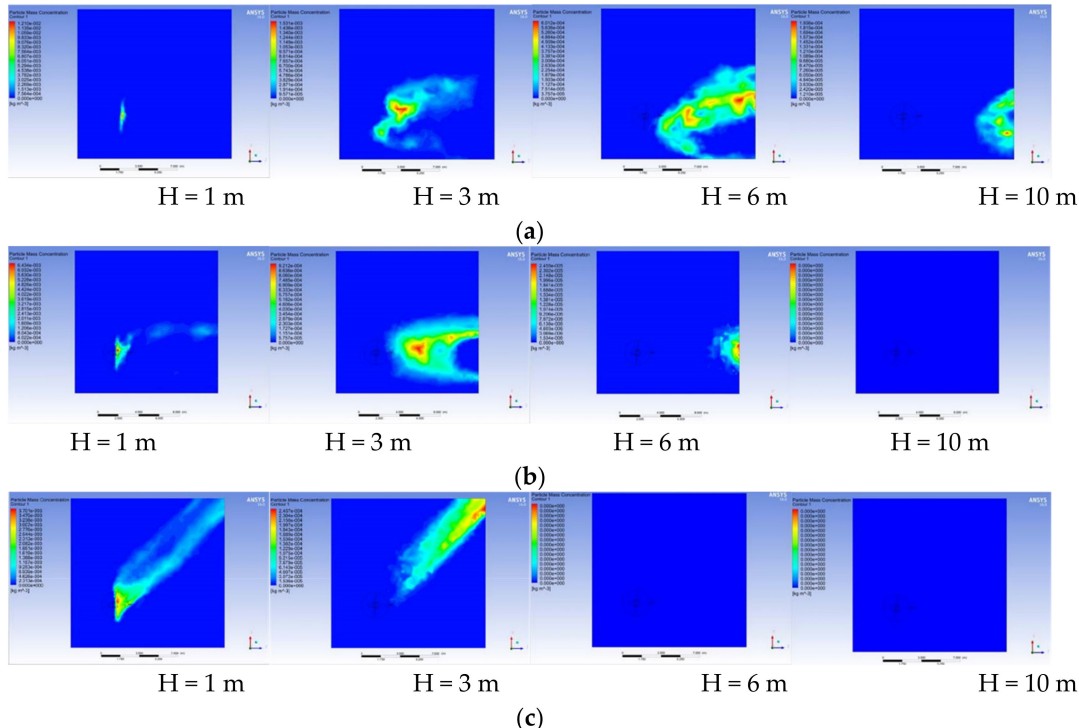

| H = 1 m | H = 3 m | H = 6 m | H = 10 m |

(a)

| H = 1 m | H = 3 m | H = 6 m | H = 10 m |

(b)

| H = 1 m | H = 3 m | H = 6 m | H = 10 m |

(c)

**Figure 9.** Cloud plot of mass concentration of droplet deposition in different height planes: (**a**) 3–0. (**b**) 6–0. (**c**) 3–3. Note: The height used for the simulation has been marked below the image. The different velocity combinations in the experiment are expressed as n-m, where n denotes the UAV flight speed and m denotes the side wind speed.

From Figure 9, it can be seen that the droplets are affected by different conditions. From Figure 9, it can be seen that the droplets are affected by different front wind, side wind, and rotor downwash wind fields in a downward spiral, and the deposition concentration of droplets at the same height varies greatly under different wind speed conditions. When the side wind is 0 m/s, only the front wind speed is changed, the front wind increases to 3 m/s at the 1 m height section, and the droplets are dispersed to the rear of the fuselage.

When the front wind is 6 m/s, the droplet deposition area gradually spreads to the tail section, and the droplet deposition concentration decreases slightly at this time. At 6 m/s front wind, the droplets spread to the rear area of the tail of the UAV and form a leakage area in the direction of the fuselage. At the 6 m height section and 10 m height section, the deposition area is disturbed, and most droplets have escaped from the calculated area when the forward wind speed is above 3 m/s. When the side wind and front wind speed are changed simultaneously, the droplet deposition shape is 45° relative to the UAV course when the side wind and front wind speeds increase to 3 m/s at 1 m and 3 m altitude sections, and no droplet deposition is observed in the calculation area when the side wind and front wind speeds increases to 3 m/s at the 6 m and 10 m altitude sections.

According to other results of the simulation, at the height of 1 m, the effect of the front wind speed in the range of 0 to 3 m/s and the side wind speed in the range of 0 to 6 m/s on the deposition concentration of droplets is relatively low. The droplets in the nozzle leave the dominant droplet trajectory. The main influencing factor is the 3 m height plane ambient wind speed. When the front wind and side wind speeds are less than 3 m/s, droplets produce drift, the deposition distribution is uniform, and there are no wind state droplets deposited under the UAV. When the wind speed increases, the deposition distribution area increases, the position of the UAV is adjusted according to the size and direction of the ambient wind speed during the actual spraying work, and the deposition quality is poor when the front wind speed is greater than 3 m/s or the side wind speed is greater than 3 m/s. At 6 m and 10 m heights, droplets are deposited only under the condition of no ambient wind speed, and the effective spraying width of droplets is drastically reduced because the droplets are affected by the rotating and falling rotor wind field and produce an angular twist.

As shown above, the reasonable height of the 3WQF120-12 single-rotor plant protection UAV's droplet spraying operation falls within the range from 1 to 3 m. When the spraying height is less than 1 m, the droplets are concentrated and the deposition area is small, which is not conducive to the full diffusion of the droplets and leads to an excessive local droplet density. When the spraying height is greater than 3 m, the droplets are influenced by the ambient wind and the droplets are too scattered and unevenly deposited, which is not conducive to the precise action on the target. The wind speed of the working environment and the flight speed of the single-rotor plant protection UAV of type 3WQF120-12 should be less than 3 m/s, and it is difficult to control the size and area of deposition density when it is greater than 3 m/s.

### 3.3. Field Trial Validation of the 3WQF120-12 UAV Spraying Model

3.3.1. Experimental Protocol

To study the spraying mechanism of the single-rotor plant protection UAV under different conditions, Wang et al. [41] conducted an outdoor experiment to verify the droplet drift of single-rotor plant protection UAVs on a pineapple field. The experimental scheme is shown in Figure 10, with two collection strips that are 40 m apart and perpendicular to the UAV's flight direction, and the length of the strips is 60 m. The center of the route is marked with 0, and the upwind position is marked with four points at −1, −2, −3, and −4 at 1 m intervals from 0 (including 0) and three points at −6, −8, and −10 at 2 m intervals with Mylar cards. Mylar cards are placed at 1 m intervals at points 1, 2, 3, and 4 in the downwind direction from course 0. The right side of point 4 is the preset drift area, and Mylar cards are placed at 2 m intervals at points 6, 8, and 10, and 10 m intervals at points 20, 30, 40, and 50. After each UAV sortie, the sampling samples were collected with disposable gloves when the sampling device was completely dry. All of the samples were sequentially numbered and placed in ice boxes and brought back to the laboratory for analysis and processing. The water-sensitive paper was analyzed with the processing software DepositScan (USDA), and the Mylar card and polyethylene line were eluted with 20 mL of ultrapure water, which was then placed in a cuvette to measure the fluorescence of the samples to calculate the concentration of rhodamine B. The instrument

used was a molecular fluorescence spectrophotometer, model F7000 (HITACHI, Ltd., Tokyo, Japan), the fluorescence–concentration curve was fitted with a correlation of 99.9%, and the concentration value of the sample was obtained and the deposition amount of the sample was calculated according to Equation (7).

$$\beta_{dep} = \frac{\left(\rho_{smql} - \rho_{blk}\right) \times F_{cal} \times V_{dii}}{\rho_{spray} \times A_{col}} \tag{7}$$

where $\beta_{dep}$ is the drift deposition, $\mu L/cm^2$; $\rho_{smql}$ is the fluorescence spectrophotometer reading; $\rho_{blk}$ is the blank comparison; $F_{cal}$ is the calibration factor; $V_{dii}$dii is the volume of pure water used to dilute the tracer, L; $\rho_{spray}$ is the spray concentration, g/L; and $A_{col}$ is the sampling card area, $cm^2$.

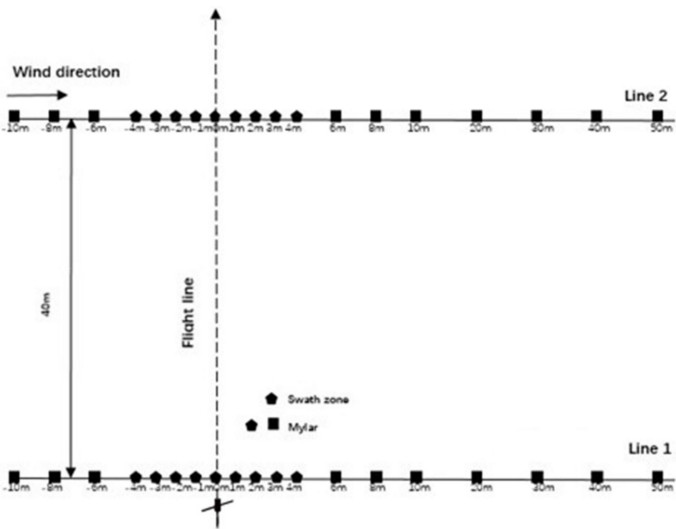

**Figure 10.** Layout of the UAV spray drift test in the field trial.

### 3.3.2. Validation of the Simulation Results of the Spray Application Model in CFD

To study the droplet deposition drift characteristics, the experimental control parameters were the operating height and ambient wind speed, and the UAV operating speed was kept at a fixed value of 3 m/s. The real-time meteorological condition parameters and the simplified abbreviations of the sorties during the specific operations are shown in Table 7, and the operating height refers to the height from the canopy. The Mylar card deposition distribution curves of a and b, c and d, and e and f in the sampling area are plotted in Figure 11.

**Table 7.** Meteorological conditions and UAV operating parameters.

| Parameters | Processing a | Processing b | Processing c | Processing d | Processing e | Processing f |
|---|---|---|---|---|---|---|
| Wind speed (m/s) | 4.7 | 1.8 | 0.7 | 2.2 | 3.7 | 1.78 |
| Wind direction (°) | 63 | 100 | 160 | 120 | 55 | 56 |
| Operating height (m) | 2.5 | 2.5 | 1.5 | 1.5 | 3.5 | 3.5 |
| Operating speed (m/s) | 3 | 3 | 3 | 3 | 3 | 3 |
| Average temperature (°C) | 27.2 | 26.1 | 27.8 | 25.9 | 24.9 | 26.5 |
| Average relative humidity (%) | 50.8 | 60.55 | 60.8 | 57.6 | 67.6 | 57.6 |

Note: The UAV was flown in six sorties, labelled a to f. This table shows the flight parameters and meteorological parameters for each sortie.

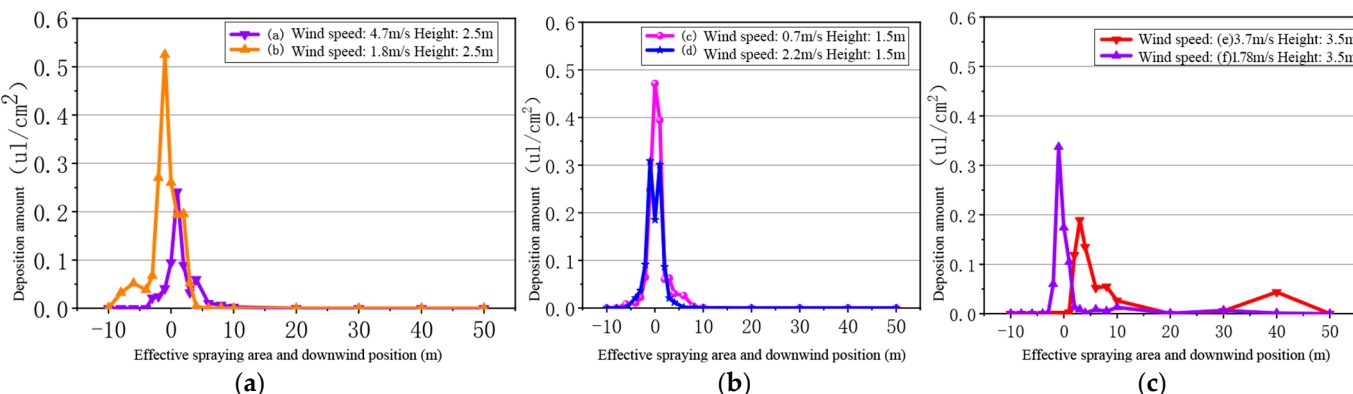

**Figure 11.** (a–f) Mylar cards in the effective spray width area and downwind deposition. (**a**) Deposition of group a and group b (group a wind speed 4.7 m/s; group b wind speed 1.8 m/s; the height of both is 2.5 m). (**b**) Deposition of group c and group d (group c wind speed 0.7 m/s; group d wind speed 2.2 m/s; the height is 1.5 m). (**c**) Deposition of group e and group f (group e wind speed 3.7 m/s; group f wind speed 1.78 m/s; the height is 3.5 m).

The spraying simulation was carried out in CFD software according to the operating parameters of the pineapple field trial, and the wind speed of the side wind was adjusted appropriately. In the comparison of Figures 11 and 12, with an operating height of 2.5 m, the side wind increases, the droplet deposition area is backward, and CFD simulations at lower wind speeds show larger deposition areas and more uniform deposition. At an operating height of 1.5 m, the simulation and the field test map of the curve and the spray area are similar, which is mainly due to the lower flight height, and the droplets are not completely dispersed. At an operating height of 3.5 m, the results are basically similar at a low wind speed. The simulated results were almost zero at a high wind speed, and the reason for this was that the field trial conducted sampling point sampling and found the influence of crop canopy structure, which led to a certain randomness of the sampling results. The different locations of the extraction sampling lines may also lead to differences in the data results, and the deposition effect of the droplets can be discussed in conjunction with the deposition concentration cloud map.

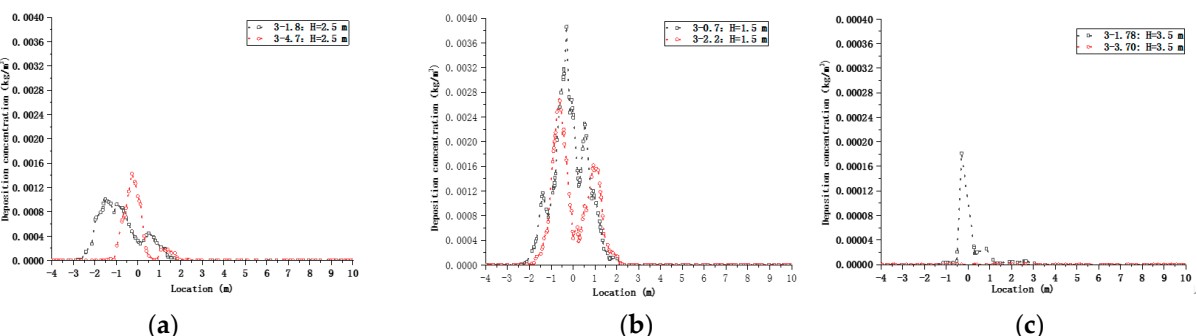

**Figure 12.** CFD simulation of the sampling line below the nozzle and droplet deposition concentration cloud. (**a**) 3–1.8 and 3–4.7 (the height is 2.5 m). (**b**) 3–0.7 and 3–2.2 (the height is 1.5 m). (**c**) 3–1.78 and 3–3.7 (the height is 3.5 m). Note: The different velocity combinations in the experiment are expressed as n-m, where n denotes the UAV flight speed and m denotes the side wind speed.

The conclusion that the operating height is not higher than 2.5 m and the wind speed is not greater than 2 m/s given by the test results was compared with the cloud chart. From Figure 13a, it can be seen that when the wind speed is lower than 1.8 m/s and the height is 2.5 m, the droplets' deposition is more concentrated, and the deposition's concentration is higher. When the side wind speed increases to 4.7 m/s, the droplet deposition area deviates from the route's position, and the deposition concentration value is nearly 10 times greater

compared with the low wind speed; moreover, the deposition area expands and easily causes drifting phenomena. When the height is 1.5 m, the side wind speed increases to 6 m/s, the droplets' deposition area becomes narrower, and the deposition concentration decreases rapidly. When the operating height is 3.5 m and when there is no side wind effect or it is below 1 m/s, the deposition area is larger, and the droplets' deposition concentration is similar. When the side wind speed gradually increases to 3 m/s, the deposition area of the droplets deviates from the UAV's heading, the droplet inhomogeneity increases, and the deposition concentration decreases several times. The combination of Figures 11–13 proves that the conclusions given in the previous paper are feasible and also verify the validity of the CFD simulation's results, providing more objective and multi-angled data to choose from.

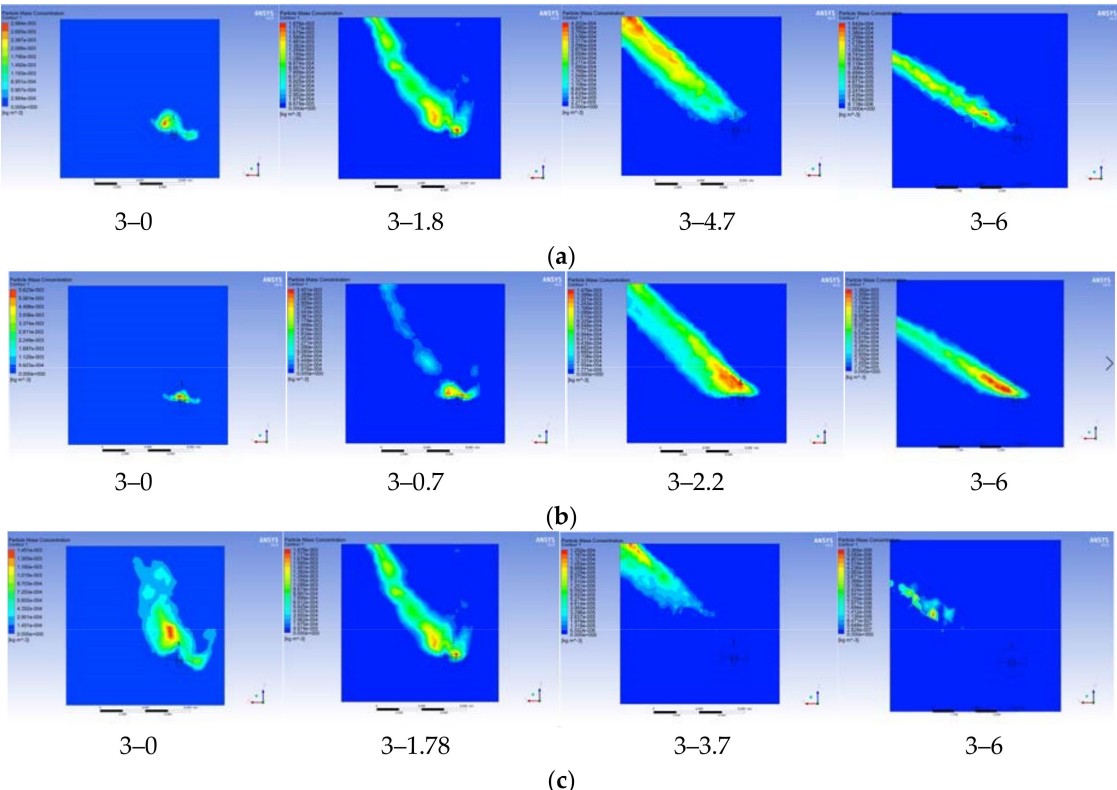

**Figure 13.** CFD simulated cloud of droplet deposition concentration for different wind speeds and heights. (**a**) H = 2.5 m. (**b**) H = 1.5 m. (**c**) H = 3.5 m. Note: The wind speed used for the simulation is marked below the picture. The different velocity combinations in the experiment are expressed as n-m, where n denotes the UAV flight speed and m denotes the side wind speed. H indicates the height of the simulation setting.

## 4. Conclusions

In this study, CFD simulations were carried out for the Lu120-015 fan nozzle commonly used in the 3WQF120-12 single-rotor plant protection UAV using computational fluid dynamics simulations. Pearson correlation analyses were performed between the experimental and simulated values, and the correlation coefficient of multiple groups reached above 0.89, which is an extremely strong correlation. For the 3WQF120-12 single-rotor plant protection UAV, the effects of the rotor wind field, flight speed, and side wind speed on the droplets' motion distribution were studied. Based on the subgrid-scale turbulence model in ANSYS Fluent, a three-dimensional virtual spray environment was established using the Euler–Lagrange method. The droplets' size and the deposition drift characteristics of the UAV at different wind speeds were analyzed. The analysis shows that side and front winds have significant effects on the droplets' motion range and particle size. This paper

provides guidance for selecting the appropriate working altitude and flight speed for this UAV. In addition, a field test verification of the UAV spraying model was conducted and simulated, and field test graphs were obtained with similar curves and similar curve spray width areas, which verified the accuracy of the CFD simulation's results.

**Author Contributions:** Writing—original draft preparation, J.W., X.L. (Xiaoyi Lv), and B.W.; investigation, X.L. (Xinguo Lan); data curation, Y.Y.; writing—review and editing, Y.L.; visualization, S.C. All authors have read and agreed to the published version of the manuscript.

**Funding:** This study was funded by the Hainan Province Science and Technology Special Fund (Grant No. ZDYF2020195), the National Natural Science Foundation of China (Grant No. 32201670), the Hainan Province Science and Technology Special Fund (Grant No. ZDYF2022XDNY273), and the Academician Lan Yubin innovation platform of Hainan Province.

**Institutional Review Board Statement:** Not applicable.

**Informed Consent Statement:** Not applicable.

**Data Availability Statement:** Not applicable.

**Acknowledgments:** The authors are very grateful to Zhizhou Deng and Along Sun for their contributions.

**Conflicts of Interest:** The authors declare no conflict of interest.

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
