# Peer review of "Numerical Simulation and Analysis of Droplet Drift Motion under Different Wind Speed Environments of Single-Rotor Plant Protection UAVs"

_drones, doi:10.3390/drones7020128_

Round 1
Reviewer 1 Report
The current state of knowledge indicates that pesticides will still continue to be used in plant protection. However, even when they will be replaced by less toxic plant protection products and natural bioproducts, they should be used sparingly due to the high costs of plant protection. There are particularly high hopes in Asian countries for the use of sprayers mounted on UAVs, but thorough research in this field does not confirm the enthusiasm for this technique, as outlined in the introduction of the article (line 32-40). The research results so far do not support the belief that UAVs are characterized by higher efficiency and lower labor consumption, as well as lower health hazards when compared to conventional tractor sprayers.
Although the subject of the article is very important for the further development of UAV spraying technique, the article in this form is not suitable for publication due to numerous deficiencies.
This belief is supported by the following the article's shortcomings.
- There is a lack of a research problems that should result from a review of existing studies and a lack of identified gaps that should be filled by the proposed research.
- The aim of the study was not presented.
- All drawings are illegible.
- The research methodology is described very briefly. There is no description of the methodology for obtaining “experimental values”.
- How was the droplet size measured? (line 353-280)?
- There is no procedure for measuring of “sedimentation amount” (fig. 10).
- Many of the terms used are not explained e.g. Lu120-015, sedimentation concentration, analog value.
- Figures captions are not understandable without using the text of the article (they should be self-explanatory).
- Company names and locations (city, country) are missing, e.g. for nozzles and measuring equipment.
- The summary does not follow from the results obtained. How is it possible to derive “the guidance for selecting the appropriate working altitude and flight speed” (line 506) from the results obtained?
Author Response
Thank you for your valuable comments on this paper. Please see the attachment.

Reviewer 2 Report
The paper presents a CFD study compared to real data of droplet motion under different winds for agricultural applications using single rotor UAV. The Introduction is well written and introduces perfectly the reader in the research. Materials and method are described and clearly expressed. Figures in this section are consistent with text. Geometric model and computational model are shown and tables state the status. Results are presented as there are strong correlation: this must be specified better. It i snot clear just looking at the Pearson coefficients. lines 346-347 and 381-383 must be rephrased...english not clear.
A lot of simulations have been preformed and the outcomes presented. Conclusions are coherent with what observed in the field and confirmed by CFD simulations.
Author Response

(The authors gave the same response as above.)

Round 2
Reviewer 1 Report
The article has been significantly improved and can be published in the presented version. Although the description of the drawings has been supplemented, some of the drawings are still illegible and may be incomprehensible to the reader. See, for example, drawings 5, 7, 8, 9